# Variations of Secondary Metabolites among Natural Populations of Sub-Antarctic *Ranunculus* Species Suggest Functional Redundancy and Versatility

**DOI:** 10.3390/plants8070234

**Published:** 2019-07-19

**Authors:** Bastien Labarrere, Andreas Prinzing, Thomas Dorey, Emeline Chesneau, Françoise Hennion

**Affiliations:** 1UMR 6553 ECOBIO, Université de Rennes 1, OSUR, CNRS, Av du Général Leclerc, F-35042 Rennes, France; 2Institut für Systematische und Evolutionäre Botanik, Zollikerstrasse 107, 8008 Zürich, Switzerland

**Keywords:** *Ranunculus biternatus*, *Ranunculus pseudotrullifolius*, *Ranunculus moseleyi*, secondary metabolite variation, amines, quercetins, natural populations, environment, redundancy, sub-Antarctic plants

## Abstract

Plants produce a high diversity of metabolites which help them sustain environmental stresses and are involved in local adaptation. However, shaped by both the genome and the environment, the patterns of variation of the metabolome in nature are difficult to decipher. Few studies have explored the relative parts of geographical region versus environment or phenotype in metabolomic variability within species and none have discussed a possible effect of the region on the correlations between metabolites and environments or phenotypes. In three sub-Antarctic *Ranunculus* species, we examined the role of region in metabolite differences and in the relationship between individual compounds and environmental conditions or phenotypic traits. Populations of three *Ranunculus* species were sampled across similar environmental gradients in two distinct geographical regions in îles Kerguelen. Two metabolite classes were studied, amines (quantified by high-performance liquid chromatography and fluorescence spectrophotometry) and flavonols (quantified by ultra-high-performance liquid chromatography with triple quadrupole mass spectrometry). Depending on regions, the same environment or the same trait may be related to different metabolites, suggesting metabolite redundancy within species. In several cases, a given metabolite showed different or even opposite relations with the same environmental condition or the same trait across the two regions, suggesting metabolite versatility within species. Our results suggest that metabolites may be functionally redundant and versatile within species, both in their response to environments and in their relation with the phenotype. These findings open new perspectives for understanding evolutionary responses of plants to environmental changes.

## 1. Introduction

Plants are sessile organisms that have to face changes in their environment. The metabolome stands at an interface between perception of environmental signals and their translation in life history traits, therefore playing a major role in allowing the organism to sustain environmental constraints. Plant metabolites were selected through long-term adaptation and diversification [1]. While primary metabolites (e.g., protein amino acids, carbohydrates), defined as essential to plant physiology are relatively few, the secondary metabolites are especially diverse [2]. Secondary metabolites play major roles in the interactions between the plant and its environment [3], being involved in protection against environmental stresses, competition, or pollinator attraction [2,4,5] and some in vegetative or floral development [6,7]. While the macroevolution of secondary metabolites is becoming more and more precisely deciphered [1,8,9,10,11,12], in contrast relatively little is known about their microevolution [13,14]. In particular, the patterns of variation of the metabolome in nature need to be further investigated.

In nature, plants form populations and in general, plants within populations are more closely related while plants among populations may show stronger evolutionary differences. Equally, populations within regions often tend to be more closely related to each other than populations among regions. Variability in the metabolite composition among populations within species is known for many plant species and metabolites [15,16,17,18,19,20,21,22]. Larger-scale variation such as among regions or latitudes was also investigated, sometimes with the aim to enrich biochemical resources. Thus, studies also show variability in plant metabolite composition across latitudinal gradients or regions [23,24]. Yet, environments may be more different among populations or regions as within, and differences in metabolite composition among populations or regions might hence be dependent or independent of environments. However, relatively few studies have examined differences in metabolite composition among populations or regions, trying to dissociate metabolite variations that are dependent on environments and such that are independent [23,24,25,26,27]. This latter part of variation in metabolite composition that would be independent of environmental factors would likely reflect either selection by unknown past environments or neutral microevolutionary differentiation among populations or regions [13,14] and remains largely to be described [9,28].

High diversity of secondary metabolites has already been reported to correlate with functional diversity [2,5,29]. First, functional convergence is observed through metabolite macroevolution [1,7]. Depending on lineage or species, different metabolites may respond to the same environment or be related to the same trait, called “functional redundancy”. Such functional redundancy is observed across all metabolites and plant lineages [1]. Yet, functional redundancy of secondary metabolites might also be observed among plants within species, but this remains to be investigated [30]. Secondly, metabolite versatility has also been observed: a given metabolite may have different roles within a given plant species in different organs or in different environments [1,31,32]. However, metabolite versatility has had little attention. Notably, whether redundancy or versatility might also emerge among populations that occupy similar environments but distant regions remains unknown. We hypothesize that, within species, neutral microevolutionary differentiation among distant but environmentally similar regions exists, and it includes the function of metabolites. Specifically, we hypothesize that functional redundancy or versatility exist within species among plant populations occupying such distant regions. A first step to test these hypotheses would be to search for patterns of correlations in nature that are consistent with functional redundancy or versatility of metabolites, paving the ground for later investigation of their physiological functions. For such a correlational approach, we predict that depending on the region, different metabolites might be correlated to the same environmental factor or the same trait in plants, suggesting functional redundancy. Moreover, depending on the region, the same metabolite might be correlated to different environmental factors or different traits in plants, suggesting functional versatility.

We aim to determine differences in secondary metabolite composition or function in natural populations of plants located in distant but environmentally similar regions. This aim requires several populations of given species distributed across wide environmental gradients within each of multiple regions. These conditions are present in sub-Antarctic Iles Kerguelen [26]. Located in the southern Indian Ocean the Iles Kerguelen harbor a wide range of abiotic environmental conditions [33] and distinct regions across which plants are distributed. This also requires metabolites that are known to vary and play roles in the response to environment or in traits. Among other metabolites, such is the case with amines and flavonoids [34,35]. Previous work comparing the amine metabolomes of nine species showed that amine composition of populations of plants in Iles Kerguelen varied in relation to both species and the environment [26]. Furthermore, previous work on flavonoids in the three *Ranunculus* species growing in Iles Kerguelen showed that quercetins were the only flavonols in these species and that composition of populations varied in relation to both species and the environment [36]. Amines include aliphatic polyamines, acetyl conjugates, and aromatic amines. Polyamines are low molecular weight polycationic molecules with amino groups [7,37] and are described as “growth regulators”, being involved in various internal processes such as growth control, DNA replication and cell differentiation, and organ development [3,34]. Polyamines, and aromatic amines by now, are also known to respond to external environment and be involved in the protection of plants against abiotic stresses [7,37,38]. Quercetins are compounds with variable phenolic structures; they belong to flavonols among flavonoids [35]. Quercetins are mainly antioxidants and are involved in the protection of plants against abiotic stresses [35]. A trade-off between flavonoid (including quercetin) concentration and growth is commonly suggested [39]. Therefore, to address the complex variation of plant metabolite composition across populations, environments, traits, and regions in nature, we performed targeted analyses of amines and quercetins. Using two independent metabolite families, we aimed at reinforcing our conclusions.

We sampled populations of the three *Ranunculus* species native to Iles Kerguelen across environmental gradients in two different geographical regions. We focused on environmental factors known to have major effects on these plants, i.e., soil hydric conditions [36,40]. We measured plant metabolite composition (amines and quercetins) and morphological phenotype, and characterized the environment (soil hydric conditions) of populations. We asked whether (i) environment alone explains differences in metabolite composition among populations within species or whether environment-independent microevolutionary differentiation among regions also shapes metabolite composition, (ii) metabolites show patterns consistent with redundancy within species, different metabolites being related to the same environments or morphological traits in the two different regions, (iii) metabolites show patterns consistent with versatility within species, the same metabolites being related to different environments or morphological traits in the two different regions.

## 2. Results

### 2.1. Populations Differ in Total Metabolite Contents and Metabolite Composition Partly Independently of the Environment

To identify an effect of population on total metabolite contents that was not due to environmental differences among populations, we conducted an ANCOVA with environmental conditions as co-variables. We found that total contents of amines or quercetins differed significantly among populations across regions and within regions in most cases, i.e., statistically independent of the environmental variables considered (Table 1 and Figure 1).

To identify an effect of population on compositions (rather than totals) of metabolites, we conducted multivariate ANCOVAS with environmental conditions as co-variables. We found that metabolite (amine or quercetin) composition differed significantly among populations across regions within species, i.e., statistically independent of the environmental variables considered (Table 2).

### 2.2. Environments Partly Explain Variation of Total Metabolite Contents and Metabolite Composition across Populations

For total amine content, relationships with environmental conditions (soil water saturation, pH or conductivity) were significant for five out of nine comparisons in *R. biternatus* (across or within regions) and three in both *R. pseudotrullifolius* and *R. moseleyi* (Appendix A).

Amine composition was significantly related to overall environment and individual environmental conditions, within or across regions in the three species (Table 3 and Figure 2 for an example of the full multivariate relationships between all environmental variables and all compounds). Testing the relationships between amine composition and individual environmental conditions, we found that individual amine levels were correlated to soil water saturation in 17 out of 48 comparisons (across or within regions) in *R. biternatus*, 16 in *R. pseudotrullifolius*, and six in *R. moseleyi* (Appendix A). Individual amine levels were correlated to soil pH in 11 out of 48 comparisons in *R. biternatus* and *R. pseudotrullifolius* and 15 in *R. moseleyi*. Individual amine levels were correlated to conductivity in 20 out of 48 comparisons in *R. biternatus*, 21 in *R. pseudotrullifolius*, and 17 in *R. moseleyi* (Appendix A).

For total quercetin contents, relationships with environmental conditions were significant for five out of nine comparisons (across or within regions) in either *R. biternatus* or *R. pseudotrullifolius* and 2 in *R. moseleyi* (Appendix A).

Quercetin composition was significantly related to overall environment in the three species (Table 3). Individual quercetin levels were significantly correlated to water saturation level in nine out of 27 comparisons (across or within regions) in *R. biternatus*, 11 in *R. pseudotrullifolius*, and 16 in *R. moseleyi* (Appendix A). Individual quercetin levels were significantly correlated to soil pH in 10 out of 27 comparisons in *R. biternatus*, 12 in *R. pseudotrullifolius*, and seven in *R. moseleyi*. Finally, they were significantly correlated to soil conductivity in 13 out of 27 comparisons in *R. biternatus*, eight in *R. pseudotrullifolius*, and six in *R. moseleyi* (Appendix A).

### 2.3. Morphological Phenotypes Partly Explain Variation of Total Metabolite Contents and Metabolite Composition across Populations

For total amine content, relations with morphological traits were significant in one out of 15 comparisons (across or within regions) in *R. biternatus*, three in *R. pseudotrullifolius*, and four in *R. moseleyi* (Appendix A).

Amine composition was significantly related to the morphological phenotype across or within regions in the three species (Table 4). Traits correlated to amine composition were mainly plant height and the number of leaves (Appendix A). Individual amine levels were correlated to plant height in nine out of 48 comparisons (across or within regions) in *R. biternatus*, 8 in *R. pseudotrullifolius*, and 11 in *R. moseleyi*. Individual amine levels were correlated to the number of leaves in four out of 48 comparisons in *R. biternatus*, six in *R. pseudotrullifolius* and four in *R. moseleyi* (Appendix A).

For total quercetin content, relations with phenotypic traits were significant in three out of 15 comparisons (across or within regions) in *R. biternatus*, three in *R. pseudotrullifolius*, and five in *R. moseleyi* (Appendix A).

Quercetin composition was significantly related to the morphological phenotype in *R. biternatus*, *R. pseudotrullifolius* and *R. moseleyi* (Table 4). Traits influenced by quercetin composition were mainly plant height and the number of leaves (Appendix A). Individual quercetin levels were significantly correlated to plant height in three out of 27 comparisons in *R. biternatus*, eight in *R. pseudotrullifolius*, and nine in *R. moseleyi*. Individual quercetin levels were significantly correlated to the number of leaves in 13 out of 27 comparisons in *R. biternatus*, nine in *R. pseudotrullifolius*, and one in *R. moseleyi* (Appendix A).

### 2.4. Regions and Metabolite-Environment Relationships

For total metabolite contents, we found 3/12 (amines) and 4/12 (quercetins) significant effects of regions on the relationships between total metabolite contents and environmental conditions (overall environment, soil water saturation, pH, conductivity) (Table 5, examples in Figure 2). These significant effects concerned *R. pseudotrullifolius* and *R. moseleyi* but not *R. biternatus*.

For amine composition, in multivariate analyses the relationship between environmental conditions and levels of individual amines was significant in one region but not in another in five out of eight comparisons within *R. biternatus* and *R. pseudotrullifolius* (Table 3). Overall, environment explained variance in amine composition better within than across regions in nine out of 12 comparisons (Table 3). In univariate analyses, the relationship between environmental conditions and levels of individual amines was significant in one region but not the other in 57 out of 144 comparisons, was significant in both but changed sign in one comparison, and was significant in both with the same sign in 11 out of 144 comparisons (Appendix A). Changes among regions were two times more frequent for soil water saturation or conductivity than for pH (24/1/1 and 21/0/7 vs. 12/0/3; respectively changes from significant to not significant/positive significance to negative/significant with same sign). Also, changes among regions were two times more frequent in *R. pseudotrullifolius* and *R. biternatus* than in *R. moseleyi* (23/1/3 and 24/0/1 vs. 10/0/7) (Appendix A). For example, in *R. biternatus*, a redundancy analysis accounting simultaneously for all multiple amines showed major shifts of relationships among regions: see, for instance, the changes in relative positions of Agm, Put, Spd and Dop (Figure 3).

For quercetin composition, in multivariate analyses the relationship between environmental conditions and levels of individual quercetins was significant in one region but not in the other in one out of 12 comparisons (Table 3). Overall, environment explained variance in quercetin composition better within than across regions in nine out of 12 comparisons (Table 3). In univariate analyses, the relationship between environmental conditions and levels of individual quercetins was significant in one region but not the other in 36 out of 81 comparisons, was significant in both but changed sign in two comparisons, and was significant in both with the same sign in eight out of 81 comparisons (Appendix A). Changes among regions were somewhat more frequent with respect to pH than with respect to soil water or conductivity (15/1/1 vs. 10/0/4 and 11/1/3) and had roughly similar relative frequency in all three species (13/0/4; 12/2/1; 11/0/3) (Appendix A).

### 2.5. Regions and Metabolite-Phenotype Relationships

For total metabolite contents, we found some 3/9 (amines) and 1/9 (quercetins) significant effects of regions on the relationships between total metabolite contents and morphological variables (Table 5). These significant effects concerned *R. pseudotrullifolius* and *R. moseleyi* but not *R. biternatus*.

For amine composition, in multivariate analyses the relationship between morphological variables and levels of individual amines was significant in one region but not in the other in three out of 12 comparisons (Table 4). Overall, morphology explained variance in amine composition better within than across regions in four out of 18 comparisons (Table 4). In univariate analyses, the relationship between morphological variables and levels of individual amines was significant in one region, but not the other in 27 comparisons, and once significant in both regions, but with opposite signs (out of 144 comparisons) (Appendix A). Changes among regions were two times more frequent for plant height than for leaf number (19/1/0 vs. 8/0/0). Changes among regions were somewhat rarer for *R. biternatus* than for either *R. pseudotrullifolius* or *R. moseleyi* (8/0/0, vs. 10/0/0 and 9/1/0) (Appendix A). Even for *R. biternatus,* a redundancy analysis accounting simultaneously for all multiple amines showed that amine/phenotype relationships are significant within one region (Isthme Bas) and non-significant across regions, with major shifts of the relationships among regions (Figure 4).

For quercetin composition, in multivariate analyses the relationship between morphological variables and levels of individual quercetins was significant in one region, but not in the other in two out of 18 comparisons (Table 4). Overall, morphology explained variance in quercetin composition better within than across regions in three out of 18 comparisons (Table 4). In univariate analyses, the relationship between morphological variables and levels of individual quercetins was significant in one region but not the other in 23 comparisons and was never significant in both regions (out of 81 comparisons) (Appendix A). Changes among regions were roughly equally frequent for plant height and for leaf number (11/0/0 vs. 12/0/0). Changes among regions were somewhat at least two times more abundant in *R. biternatus* and *R. pseudotrullifolius* than in *R. moseleyi* (9/0/0, vs. 10/0/0 and 4/1/0) (Appendix A).

## 3. Discussion

While the macroevolution of secondary metabolites is becoming more and more deciphered, their microevolution at the intraspecific level remains far more obscure [14,25]. We investigated how secondary metabolites vary at the intraspecific level, where plants are subject to both environmental constraints and neutral microevolution. We hypothesized that microevolution within species could affect metabolite composition but also metabolite functions—i.e., the way metabolites respond to the environment or interact with the phenotype. In a correlational multi-species field study, we investigated two secondary metabolite families (amines and quercetins) among natural populations in sub-Antarctic Iles Kerguelen. Populations were distributed across two distinct regions, each region having similar environmental gradients. This pattern allows determining differences between regions in environment-metabolite or trait-metabolite relationships. Amine and quercetin compositions found in the three species were consistent with previous work [26,36]. We know from respectively Hennion et al. [26] and Hennion et al. [36] that amine or quercetin compositions in natural populations of *Ranunculus* species from Iles Kerguelen differ among environments and species. We found that variation of amine or quercetin composition among populations or regions within species is not only shaped by the environment, suggesting that either neutral microevolution or evolution under past environmental selection pressures may also shape metabolite composition within species. Moreover, we found that depending on regions, different metabolites may be correlated to the same environmental condition or the same trait, which is consistent with the hypothesis of metabolite redundancy within species. Also, depending on regions, a given metabolite may show different or even opposite correlations, which supports the hypothesis of metabolite versatility within species. Our results suggest that microevolution within species shapes secondary metabolite composition but also influences metabolite functions through metabolite redundancy and versatility.

### 3.1. Environments and Morphological Traits Partly Explained Variation of Total Metabolite Contents and Metabolite Composition across Populations

For all three species, the highest number of correlations of individual amine levels was with soil or water conductivity (Table 3 and Appendix A). In *R. biternatus* and *R. pseudotrullifolius* many correlations of individual amine levels were found with soil water saturation, more than with pH, with the reverse case in *R. moseleyi*. These findings are consistent with the known involvement of amines in plant response to water and salinity, some having demonstrated protective roles against these stresses [37]. In contrast, relations with soil pH are less known.

Regarding quercetins, there were about as many correlations of individual quercetin levels with environmental conditions as with amines, invoking equally all three environmental conditions (Table 3 and Appendix A). Quercetins are involved in plant response to stress [35], however, to our knowledge, our study is the first that raised correlations between individual environmental conditions and quercetin composition of plants in nature. Our results suggest a possible involvement of quercetins in plant response to soil water saturation, conductivity and pH.

There were relatively few significant correlations of individual compounds (amines or quercetins) with morphological traits (except for quercetins with the number of leaves) (Table 4, Appendix A). Whatsoever, our study demonstrated that amine or quercetin compositions are related to both environmental conditions and traits in nature within species, a topic rarely addressed for amines [41] and never yet for quercetins [39].

### 3.2. Environment Did Not Explain All Differences in Total Metabolite Contents and Metabolite Composition among Populations

Only a few authors differentiate metabolite variation that is environment-dependent and metabolite variation that is environment-independent [13,25,26]. Here, we showed that the different populations within species share the same pool of compounds but differ quantitatively in the total contents and in the composition in amines or quercetins. Moreover, we showed that such differences in amine or quercetin totals and compositions among populations were only partly explained by the environment we had measured (Table 1 and Table 2, Figure 1). Our results might possibly suggest that we have left out important environmental variables. However, we consider this unlikely given that our previous work supports the soil hydric variables we used as discriminant for the compositions in respectively amines [26] and quercetins [26,36] of our study species. More likely, our results suggest that, to some degree, present differentiation of amines and quercetins within plant species is shaped either by adaptation to unknown past environments (not correlated to present environments), or by neutral microevolutionary differentiation among populations.

### 3.3. Differences in Relationship of Metabolites with Environment or Traits between Regions; Patterns Consistent with Metabolite Redundancy

Often, in the two different regions, the same environmental condition correlated to different compounds among amines or among quercetins. However, this did not apply to the amine composition in *R. moseleyi*: in this species, the amines that correlated to pH or conductivity were the same (four compounds) across the two regions (Appendix A).

The change in correlation significance from one region to the other was also observed for morphological traits, however here, the numbers of correlations were far less than with individual environmental conditions and mainly concerned one region (Isthme Bas) (Appendix A). As a result, there were fewer examples of “alternative” compounds correlated to the same trait across the two regions than there were when considering environmental conditions.

Whereas all three *Ranunculus* species studied displayed changes of correlations between the two regions, some differences were observed among the species. *R. pseudotrullifolius* and *R. moseleyi* showed effects of regions on the relations between the total contents of amines or quercetins and environments or traits, but not *R. biternatus* (Table 5). In contrast, at least for environmental conditions, changes in correlations of individual amine or quercetin levels with individual conditions were more frequent in *R. biternatus* and *R. pseudotrullifolius* than in *R. moseleyi* (Table 3, Figure 3, Appendix A) (whereas correlations with traits were few—Table 4, Figure 4, Appendix A). The latter result may link to the fact that the changes in correlations among regions most frequently concerned soil water saturation and conductivity, two factors less variable in the aquatic biotopes of *R. moseleyi*. Clearly, the species-specific variation patterns deserve further investigation.

Amines are known to respond to environmental stresses (reviewed in [37]). Such is also true of quercetins [35]. So, raising correlations of these compounds to environmental conditions is not surprising. In contrast, the finding that the same environmental condition correlates to different compounds across different regions is, to our knowledge, reported for the first time. If we consider the studied compounds as functional, it means that the same function in plants growing in different regions may be ensured by different metabolites. This is the definition of functional redundancy. We found this pattern in two metabolite families (amines and quercetins), and we found it among metabolites within species. Such possible metabolite redundancy among plants within species has been studied very little so far [28]. This concept has been initially defined at the interspecific level. Groppa and Benavides [3] reported that across studies different secondary metabolites responded to the same environments among different species—even if many authors emphasized that different studies are often hardly comparable.

Within each of the amine and quercetin families, the compounds share metabolic pathways and may derive one from each other [7,39]. Moreover, metabolites belonging to the same family often show similar, or at least overlapping, functions in plants [7,37,39]. The proximity in synthesis pathway might trigger redundancy in function in respective amines and quercetins.

Our results strongly suggest the occurrence of functional redundancy of metabolites within species. We see an interesting issue that deserves further investigation. All the metabolites are found in all the populations studied across regions: hence, differences in metabolite-environment or metabolite-trait relationships between regions are not due to the presence or absence of different metabolites between regions. In contrast, each population potentially synthesizes several compounds that may not be used to respond to environment or interact with traits. Producing unused compounds may have a cost to plants and costs and benefits of metabolite redundancy remains to be understood.

### 3.4. Differences in Relationship of Metabolites with Environment or Traits between Regions; Patterns Consistent with Metabolite Versatility

In some cases, the same compound related to different environmental variables or morphological traits in different regions, or a given relationship changed sign (Appendix A). Secondary metabolites may be involved in different functions in different organs or in different environments within the same plant species, i.e., metabolite versatility [5,31,32]. Here we expand the concept of metabolite versatility by suggesting that the same metabolites may have different functions, and even have opposite functions facing the same environment, in different regions. Due to different or opposite effects of the same metabolite in different regions, an analysis across regions comes to the false conclusion that this metabolite likely has no effect at all. Overall, we suggest to take into account versatility in analyzing metabolite functions. The capacity of a metabolite to have different roles increases the capacity of a species to respond to the environment.

### 3.5. Metabolite Redundancy and Versatility as a Result of Microevolution Driven by Distance rather than Environment

In both regions we sampled roughly the same environmental gradients. Hence, we may probably exclude environmental differences as a cause of pattern of metabolite redundancy and versatility. Alternatively, such patterns might reflect either different metabolite plasticity or different heritable adaptations among populations within the different regions. We do have indication from previous work that some part of the amine composition of plants in natural populations is heritable or persistent over a few years [26,37].

If partly heritable metabolites relate differently to the same environments or traits in two spatially distant but environmentally similar regions, this suggests neutral microevolution between regions. In theory, such differences might also reflect adaptive microevolution under selection pressures relating to past environmental conditions, although general environmental conditions have persisted in Kerguelen since the last glacial maximum [42]. Future research should investigate whether differences of metabolite/environment/phenotype relationships between regions are heritable. Note that the three studied *Ranunculus* species exhibit frequent vegetative reproduction [43] and self-pollination, with even cleistogamy in submerged flowers in *R. pseudotrullifolius* and *R. moseleyi* [44]. Self-pollination and vegetative reproduction within populations may reduce gene flows across populations and may therefore increase microevolutionary differentiation among populations, possibly facilitating the origin of redundancy and versatility.

## 4. Materials and Methods

### 4.1. Plant Collection

The Iles Kerguelen (49°20′00″ S, 69°20′00″ E) are located in the Southern Indian Ocean within the sub-Antarctic region [45] (Figure 5). These islands are characterized by permanently low temperature (4.6 °C annual mean), strong and permanent winds (10 m·s^−1^ annual mean), and high precipitation (60 years ago but drastically reduced in recent years) (annual mean of 760 mm in the studied regions) [46].

We studied three *Ranunculus* species (i.e., *R. biternatus, R. pseudotrullifolius* and *R. moseleyi*), of which *R. biternatus* is austral circumpolar, *R. pseudotrullifolius* magellanic and on Kerguelen, and *R. moseleyi* is a strict Kerguelen endemic [47]. All are perennial plants. On the Iles Kerguelen, these species have different ecological amplitudes but occupy partially overlapping habitats [43]. *Ranunculus biternatus* is widespread on the island occurring in habitats up to 500 m above sea level. *R. pseudotrullifolius* and *R. moseleyi* have more restricted distributions. The first one, being halophilous, occurs within a short distance of the coast, occupying peaty or sandy shorelines and ponds [43]. *Ranunculus moseleyi* is strictly aquatic, growing only in freshwater lakes and ponds [43].

Plants were sampled in 18 populations (six populations per species) equally distributed across two regions in Iles Kerguelen: Isthme Bas, a large flat isthmus (about 30 km^2^) and Ile Australia, a large island (about 20 km^2^) (Figure 3). Plants across the two regions were sampled at similar altitudes and subject to similar temperatures and across the same vegetation types (herbfields, shoreline and pond vegetation) [43]. Each population was subdivided into two sites of contrasting humidity conditions and in each a continuous group of plants living in a same site was sampled. Sampling was performed across a small area (between 0.5 and 5 square meters) so as to ensure that plants assigned to a same population were subject to roughly similar environmental conditions.

The entire sampling was performed in summer during a short period, 6 weeks from mid-December 2011 to mid-March 2012. Plants were sampled between 11 h and 17 h to avoid bias from daily variation of metabolism [48,49]. An average of five individual plants per population, of the most frequent size in the local population were sampled. As the three *Ranunculus* species propagate vegetatively via runners, to avoid pseudoreplication we sampled plants at a distance above average runner length one from each other [50]. In each individual we performed morphological measurements and collected two to four leaves to quantify amines and quercetins. As metabolite composition may vary within a given plant at a given moment between leaves of different developmental stages [49,51], we sampled an appropriate and constant leaf developmental stage, i.e., fully developed photosynthetic leaves as in previous work [52]. We moreover verified whether inclusion of sampling month into an ANOVA relating region to the different compounds reduces the number of significant relationships between region and compounds and found that this is not the case. For amines, even the identity of the affected compounds remained the same in 15 out of 20 cases. The samples collected were frozen in liquid nitrogen and stored at −80 °C then lyophilized and ground to powder using a mixer mill MM 400 Retsch (Haan, Germany).

### 4.2. Plant Measurements

In each individual we measured plant height, the length and width of the largest leaf, the numbers of leaves and flowers, the flowering stage and the size of the largest flower (largest diameter). Flowering stage of the individuals was estimated following Hennion et al. [26]. To determine leaf dry matter content (LDMC), in each population a total of 20 leaves were sampled from a minimum of 15 individuals and processed following Cornelissen et al. [53]. Leaves were collected then directly put in distilled water in airtight bags for rehydration. They were then weighed before and after 48 h drying at 80 °C. Some traits (flowering stage, number of flowers and the size of largest flower) were highly redundant. We thus only kept the continuously varying trait “size of largest flower” as floral trait. Likewise, we found low variability of LDMC among populations; hence we did not use this trait in our analysis.

### 4.3. Determination and Quantitation of Free Amines and Acetylated Polyamines

This determination followed Hennion et al. [26], with modified quantities as follows. Samples from individual plants were analyzed individually. Several samples from the same population and environmental conditions were pooled in case of insufficient material (see Appendix A). Five to ten milligrams of powdered samples were thoroughly mixed with 100 to 200 μL of 1 mmol·L^−1^ HCl supplemented with 10 μmol·L^−1^ diaminoheptane (Sigma, St. Louis, MO, USA), as an internal standard, on a magnetic stirring plate (2000 rpm) for 1 h at 4 °C. The homogenates were then centrifuged for 15 min at 10,000× *g* at 4 °C, and the supernatant of each sample collected. The pellet of each sample was further extracted twice with 100 to 200 μL of 1 mmol·L^−1^ HCL and 10 μmol·L^−1^ diaminoheptane. After a short stirring period, the homogenates were centrifuged for 15 min at 10,000× *g* at 4 °C. For each sample, the three supernatants were combined and used as the crude extracts for characterization and determination of free and acetylated amines and polyamines and stored frozen at −20 °C before chromatographic analyses. High Performance Liquid Chromatography (HPLC) and fluorescence spectrophotometry were used to separate and quantify amines prepared as their dansyl derivatives according to Smith and Davies [54] with some modifications as follows. Aliquots (200 μL) of the supernatant were added to 200 μL of saturated sodium carbonate and 600 μL of dansyl chloride in acetone (7.5 mg·mL^−1^) in a 5 mL tapered reaction vial. After a brief vortexing, the mixture was incubated in darkness at room temperature for 16 h. Excess dansyl chloride was converted to dansylproline by 30 min incubation after adding 300 μL (150 mg·mL^−1^) of proline. Dansylated amines were extracted in 1 mL ethylacetate. The organic phase was collected then evaporated to dryness, and the residue was dissolved in methanol and stored in glass vials at −20 °C. External standards were made for agmatine (Agm), diaminopropane (DAP), putrescine (Put), cadaverine (Cad), spermidine (Spd), spermine (Spm), *N*-acetylputrescine (NAc-Put), *N*^8^-acetylspermidine (N^8^Ac-Spd), and *N*^1^-acetylspermine (N^1^Ac-Spm) for aliphatic amines and their acetylated conjugates; phenylethylamine (Phe), octopamine (Oct), 3-methoxy-4-hydroxy phenylethylamine (3M4OHPhe), tyramine (Tyr), and dopamine (Dop) for phenylalkylamines; tryptamine (Try) and serotonin (Ser) for indolalkylamines (all authentic products from Sigma, St. Louis, MO, USA). These standards were processed in the same way as samples, and 2 to 50 nmol (per assay) were dansylated for each standard alone or in combination. One standard combined these 15 amines plus diaminoheptane. The HPLC column was packed with reverse-phase SpherisorbODS2 C18 (particle size 5 μm; 4.6 × 250 mm, Waters, Milford, CT, USA). The mobile phase consisted of a solution of 17.5 mmol·L^−1^ potassium acetate (pH 7.17) as eluent A and acetonitrile as eluent B. The solvent gradient, modified according to Hayman et al. [55] was as described by Jubault et al. [56]. The flow rate of the mobile phase was 1.5 mL·min^−1^. For fluorescence detection of dansyl amines, an excitation wavelength of 366 nm was used with an emission wavelength of 490 nm. The external standards were injected in the HPLC system first to determine retention times of the various amines on the column and secondly to make calibration curves for quantitation. Peaks of amines in the samples were determined by their retention times on the column, and stability was checked by injection of the combined 16-amine standard in the system every 15 samples. In case of doubt, identities of peaks of amines were confirmed by spiking the sample with known amounts of the authentic standards. Amines in the samples were quantified after yield correction with the internal standard and calibration with the external standards. The stability of quantitative calibration was checked by injection of a Put standard every 10 samples. The HPLC design consisted of a thermoelectron pump (SpectraSystem P1000 XR, Thermo Fisher, San Jose, CA, USA) and (Spectra-Series AS100) autosampler with a 20 μL injection loop, and detection through an FP-2020 Plus fluorometer (Jasco, Inc., Easton, MD, USA). Signals were computed and analyzed using Azur software (Datalys, St Martin d’Hères, France).

### 4.4. Determination and Quantitation of Quercetins

Quercetins were the sole flavonols detected in *Ranunculus* species from Iles Kerguelen [36]. Samples from individual plants were analyzed individually. Several samples from the same population and environmental conditions were pooled in case of insufficient material (see Appendix A). We weighed about 10 mg of plant powder in an Eppendorf tube, and added 1 mL of methanol acidified with 1% formic acid. The tube was vortex-agitated first and put in ultra-sonic bath for 5 min. The tube was then centrifuged briefly and 900 µL of the supernatant was removed using a 1 mL plastic syringe, and filtered using a PTFE 13 mm 0.45 µm syringe filter. The methanol extract was then poured in an injection vial for Ultra Performance Liquid Chromatography (UPLC) analysis; 2 µL of the extract were injected in the Waters UPLC_PDA_ESI_TQD system for flavonol quantitation. The reversed phase column, an Acquity Waters C18 BEH (2.1 × 150 mm) 1.7 µm, was maintained at 30 °C. The solvents used for the binary gradient were A: ultra-pure water with 0.1% formic acid, B: acetonitrile with 0.1% formic, the flow was 0.4 mL/min. The gradient applied was 98% A from 0 to 0.2 min, 10% A from 0.2 to 14 min, 14 to 15 min isocratic 10% A, 15 min to 17 min 98% A, 17 to 20 min isocratic 98% A. The photo diode array detector scanned from 190 to 600 nm and flavonols were detected at 350 nm, external quantitation with some flavonol standards was applied. The UPLC-photodiode array-electrospray-triple quadrupole analytical system allows us to detect compounds for which the molecular ion produced in the electrospray source is in accordance with the molecular structure searched. The capillary voltage was 2.9 kV, the cone voltage was 37 V, the source temperature was maintained at 150 °C and the desolvatation temperature at 400 °C, the desolvatation gas flow was 800 L/h. On the basis of data from the literature on Antarctic *Ranunculus* flavonols, Gluchoff-Fiasson et al. [36], the mass spectrometer detector was programmed to focus on characteristic *m*/*z* of those flavonols yet identified. In negative mode, the ions, monitored by the Select Ion Recording method (SIR) for quantification were: 787 (Quercetin-tri-glucoside), 949 (Quercetin-caffeoyl-tri-glucoside), 933 (Quercetin-feruloyl-di-glucoside-pentoside), 919 (Quercetin-caffeoyl di-glucoside-pentoside), 757 (Quercetin-di-glucoside-pentoside), 595 (Quercetin-glucoside-pentoside), 625 (Quercetin-di-glucoside), 963 (Quercetin-feruloyl-tri-glucoside) and the standards 463 (isoquercitrin), 609 (rutin). Depending on compound structure, isoquercitrin or rutin were used as standards for the quantification on the mass spectrometer triple quadrupole detector, the external standard calibration was made daily and linear regression factors were at least 0.99.

### 4.5. Amines Characterized

We characterized 15 different amines which belonged to four biochemical categories: aliphatic amines and their acetylated conjugates, phenylalkylamines and indolalkylamines. The detected aliphatic amines were: agmatine (Agm), diaminopropane (DAP), putrescine (Put), cadaverine (Cad), spermidine (Spd), spermine (Spm), *N*^8^-acetylspermidine (N^8^Ac-Spd) and *N*^1^-acetylspermine (N^1^Ac-Spm). Phenylalkylamines were phenylethylamine (Phe), octopamine (Oct), 3-methoxy-4-hydroxy phenylethylamine (3M4OHPhe), tyramine (Tyr), and dopamine (Dop). Indolalkylamines were tryptamine (Try) and serotonin (Ser). The raw data of individual compositions are shown in Appendix A. All 15 compounds described were present in the three species, in the two regions. Thus, differences in amine composition between species or regions reflected shifts in levels and not qualitative differences.

### 4.6. Quercetins Characterized

Following Hennion et al. [36], we performed an analysis of flavonols. Quercetins characterized were: quercetin 3-diglucoside-7-glucoside (Q-3GL), quercetin 3-(caffeyl-glucosyl)glucoside-7-glucoside (Q-3GL+Caf), quercetin 3-(ferulyl-glucosyl)glucoside-7-glucoside (Q-3GL+Fer), quercetin 3-(caffeyl-xylosyl)glucoside-7-glucoside (Q-2GL+Xyl+Caf), quercetin 3-(ferulyl-xylosyl)glucoside-7-glucoside (Q-2GL+Xyl+Fer), quercetin 3-xylosylglucoside-7-glucoside (Q-2GL+Xyl), quercetin 3-xylosylglucoside (Q-GL+Xyl), quercetin 3-diglucoside (Q-2GL) and isoquercitrin (IQC). The raw data of individual compositions are shown in Appendix A. All nine compounds described were present in the three species, in the two regions. Thus, differences in quercetin composition between species or regions reflected shifts in levels and not qualitative differences.

### 4.7. Environmental Measurements

In each population we measured soil water saturation, pH and conductivity. Three samples of soil, each of 20 mL, were collected at the rhizosphere level of the measured plants. To determine soil water saturation, half of each soil sample was dried at 105 °C during 48 h and weighed before and after drying [50]. Soil water saturation in each population was calculated as following: soil water saturation = (soil weight before drying – soil weight after drying) / sol weight after drying. Data were transformed following f(x)=log(x) to reduce positive skewness of the data distribution. The remaining soil was mixed with known volume of distilled water and was then left 18 to 24 h to permit sedimentation of soil particles. Immediately after, pH was determined using a pH meter (BASIC 20 PLUS CRISON, resolution 0.01 pH). After another 18 to 24 h sedimentation, conductivity was determined using a conductivity meter (CONSORT K810, resolution 0.1 µS cm^−1^) [50].

### 4.8. Statistical Analyses

To determine differences of *total contents* of metabolites (amines or quercetins) among populations that are not due to differences in the environmental variables, we conducted ANCOVA analyses with the environment as a co-variable. We analyzed the difference between populations, accounting for environmental conditions: Total amine content in sample = pH (continuous) + Conductivity (continuous) + Water saturation (continuous) + population (six categories) + error. We consistently repeated this procedure across all populations and across only the populations within a given region to explore whether populations are more different in one region or another or whether populations vary only across regions, and not between. For this and all following analyses we used R 3.5.0 software [57]. To determine differences of total metabolite contents (amines or quercetins) among populations that may be due to differences in the environmental variables we measured, we regressed total metabolite levels against environmental predictors. We repeated this analysis separately within each of the two regions. Equally, to determine differences of total metabolite contents (amines or quercetins) among populations that relate to morphological phenotypes, we regressed total metabolite levels against morphological predictors. We repeated this analysis separately within each of the two regions.

To determine the relationships between *compositions* of metabolites (amines or flavonoids) and environment or phenotype we performed redundancy analyses, using cca function [58,59] in R 3.5.0 software [57]. We determined relationships between metabolite composition and individual environmental factors or individual traits using redundancy analyses with rda function. Also, we determined relationships between metabolite composition and the overall environment (i.e., taking into account all the environmental factors) or the overall phenotype (i.e., taking into account all the traits) using redundancy analyses. We repeated this analysis separately within each of the two regions.

To determine whether relationships between *compositions* of metabolites and the environment or metabolite-phenotype relationships differ between regions, we conducted multiple regression analyses. Dependent variables were metabolite compounds (either amines or quercetins) and independent variables were either of the environmental variables, region and the interaction term between both. We also used an integrative “overall environment” variable calculated as the scores along the first axis of a PCA across the individual environmental variables. This axis was most strongly correlated to water saturation in *R. biternatus*, and to pH in the two other species. To statistically test whether relationships between *compositions* of metabolites and the morphological phenotypes differ between regions, we took the same approach as for environment, replacing environmental by morphological variables. Again, we identified an integrative “overall phenotype” variable calculated as the scores along the first axis of a PCA across the individual morphological variables. This axis was most strongly correlated to plant height in all three species and to leaf number in *R. biternatus*, and to LDMC in the other two species.

For multiple testing on the same data set, *p*-values were corrected using sequential Bonferroni’s correction [60]. In all regression analyses we verified the assumptions of the analyses using QQ plots and predicted-vs-residual plots.

## 5. Conclusions

Recent authors [13,14] encouraged researchers to explore the relative contributions of genetic, environmental, microenvironmental and stochastic variation to secondary metabolite variation across plant taxa and environments. Our study provides several hints into secondary metabolite variation and microevolution within species. For two metabolite classes, we showed that variation of secondary metabolite composition among populations was only partly related to environment, suggesting that neutral microevolution also shapes metabolite composition within species. We showed differences in metabolite-environment and metabolite-trait relations among regions. The observed variation patterns may be interpreted as metabolite redundancy and versatility within species. Our results suggest that such possible metabolite redundancy and versatility may be shaped by neutral microevolution. Metabolite redundancy and versatility within species may contribute to the high functional diversity of individual secondary metabolites. We found patterns suggesting metabolite redundancy and versatility in three species and in two distinct families of secondary metabolites (i.e., amines and quercetins), relating to all environmental parameters and all morphological variables. Therefore, our observations likely do not result from a special case but may be extendable to other species, secondary metabolite families or locations. Future aims may be to assess the extent of metabolite redundancy and versatility, test the functions of metabolites in nature, and look at these processes in the light of costs and benefits for plant species.

## Figures and Tables

**Figure 1 plants-08-00234-f001:**
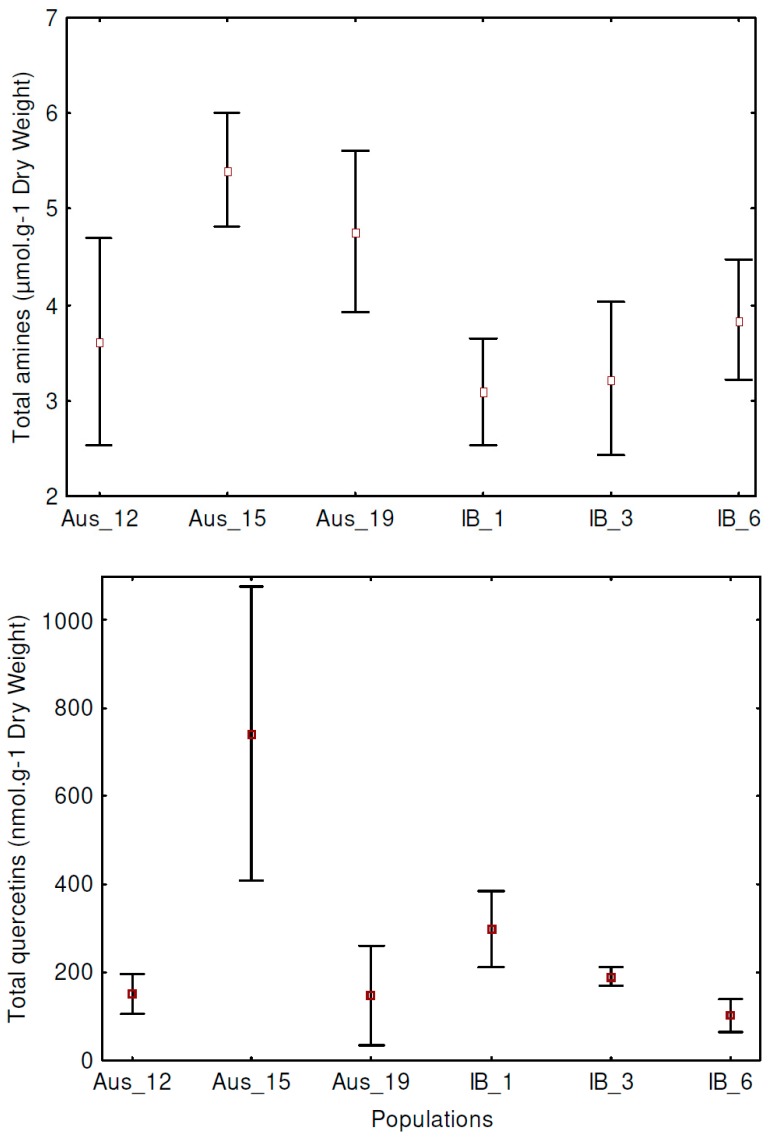
Variance of total contents of amines or quercetins among populations. Means and SD (standard deviation) are given. See Table 1 for test statistics.

**Figure 2 plants-08-00234-f002:**
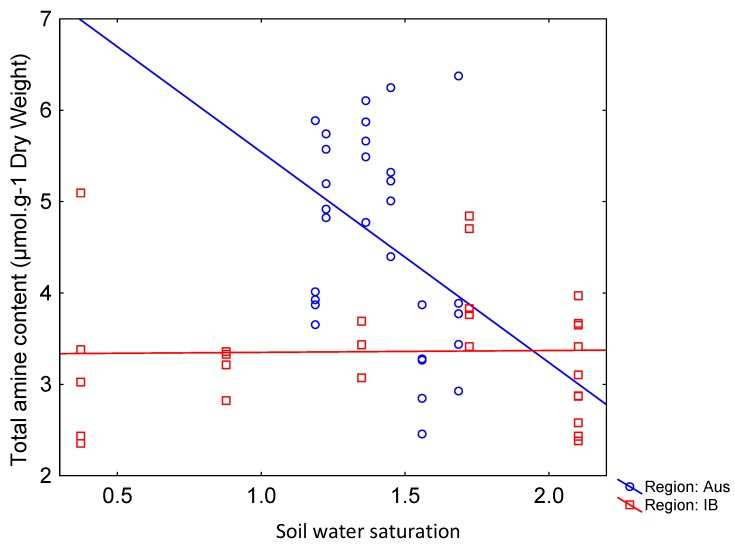
Illustration of region-dependent relationships of total amines to soil water saturation for *R. pseudotrullifolius*.

**Figure 3 plants-08-00234-f003:**
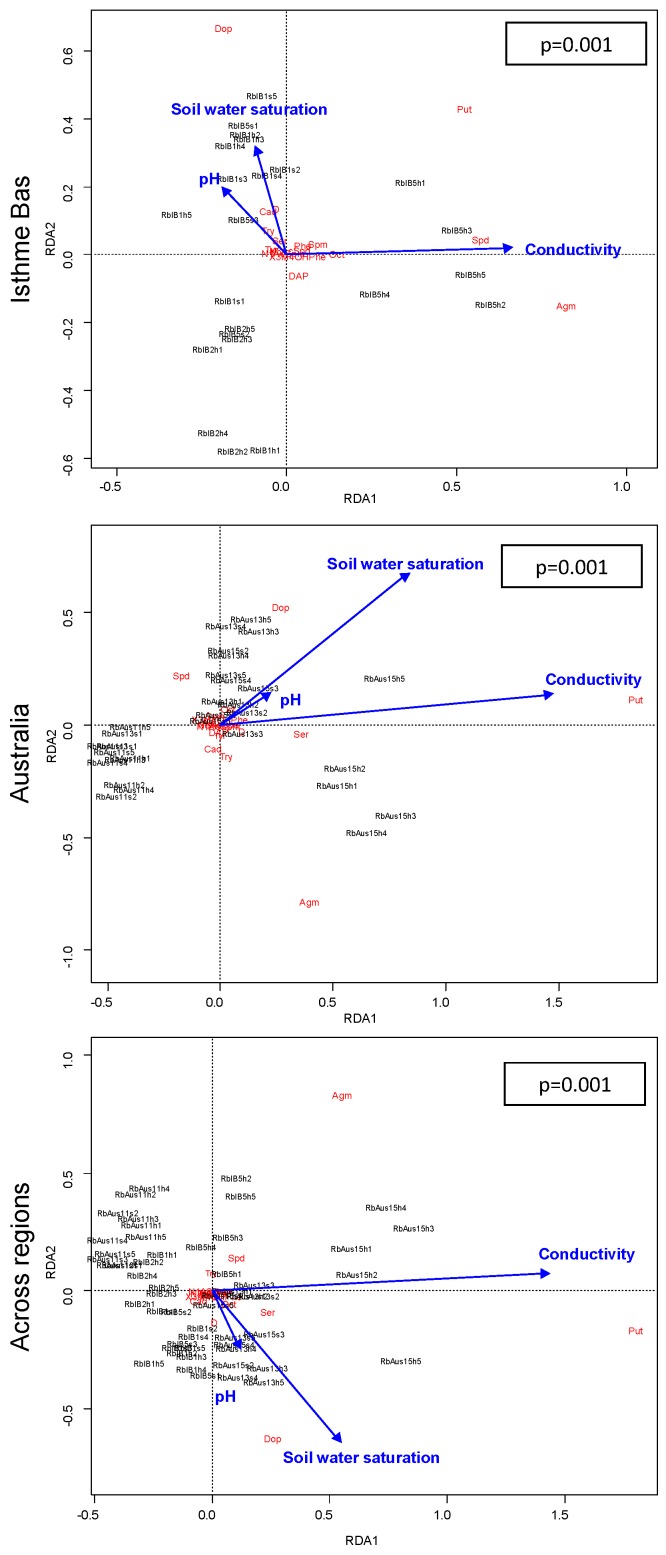
Full relationship between all environmental variables and all amine compounds, within and across regions, exemplified for *R. biternatus*. Environmental variables are soil water saturation, pH and conductivity. In redundancy analyses, axes are constrained to show, at best, variance explained by environmental variables. Individual, amines (in bold) and environmental variables (with arrows) are indicated. N = are 28 in Isthme Bas; 30 in Australia; 58 across regions. *p* values (*p*) are indicated. R^2^ are 0.49, 0.73, ad 0.54 for Isthme Bas, Australia, and across-region, respectively. Abbreviation: Agm: Agmatine; Put: Putrescine; Spm: Spermine; Spd: Spermidine; Cad: Cadaverine; DAP: 1,3-diaminopropane; Dop: Dopamine; Ser: Serotonin; Tyr: Tyramine; Oct: Octopamine; N1Ac-Spm: N^1^-acetylspermine; N8Ac-Spd: N^8^-acetylspermidine; Try: Tryptamine; 3M4OHPhe: 3-methoxy-4-hydroxy phenylethylamine (see Table 3, Appendix A for more detailed analysis, for the full set of species, and for both amines and quercetins).

**Figure 4 plants-08-00234-f004:**
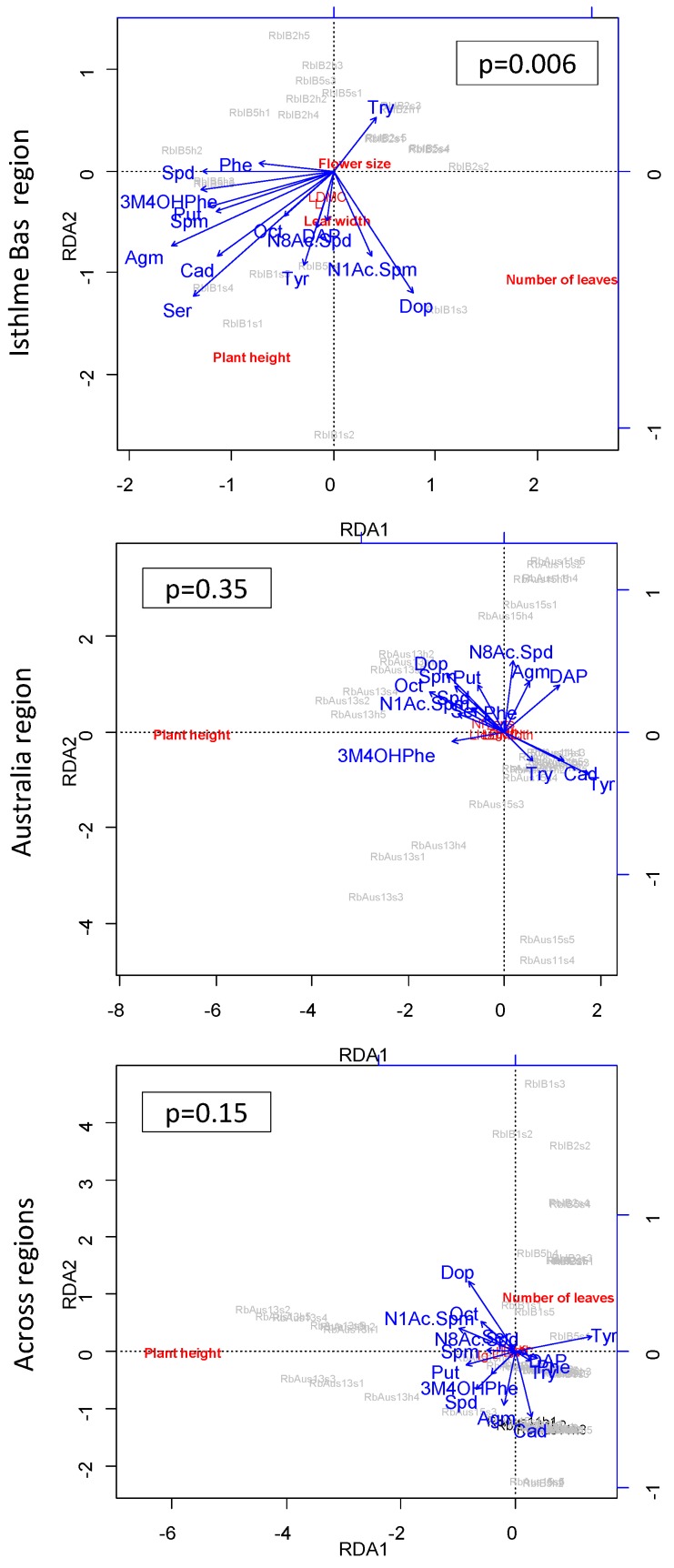
Full relationship between all amine compounds and all traits in *R. biternatus* within and across regions, from redundancy analysis. Traits are Plant height, Number of leaves, Flower size, Leaf length, Leaf width, LDMC (Leaf Dry Matter Content). In redundancy analyses, axes are constrained to show, at best, variance explained by amines. Individual traits (in bold) and amines (with arrows) are indicated. N = are 28 in Isthme Bas; 30 in Australia; 58 across regions. *p* values (*p*) are indicated. Abbreviation: Agm: Agmatine; Put: Putrescine; Spm: Spermine; Spd: Spermidine; Cad: Cadaverine; DAP: 1,3-diaminopropane; Dop: Dopamine; Ser: Serotonin; Tyr: Tyramine; Oct: Octopamine; N1Ac.Spm: N^1^-acetylspermine; N8Ac.Spd: N^8^-acetylspermidine; Try: Tryptamine; 3M4OHPhe: 3-methoxy-4-hydroxy phenylethylamine (see Table 4, Appendix A for more detailed analysis, for the full set of species, and for both amines and quercetins).

**Figure 5 plants-08-00234-f005:**
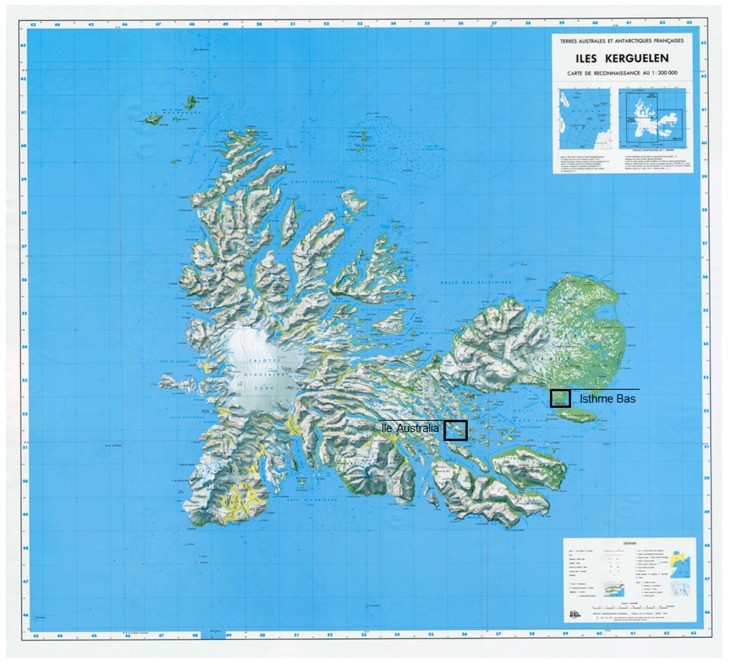
Sampled regions in Iles Kerguelen (modified from IGN [Institut Géographique National, France] map).

**Table 1 plants-08-00234-t001:** Differences in the total contents of amines or quercetins among populations *.

**Amines**		**Population**		
		***p*-Value**	***F* Value**	**df Residuals**
*R. biternatus*	across regions	**0.001**	4.794	44
	Isthme Bas	**0.004**	10.773	18
	Australia	**0.010**	6.565	24
*R. pseudotrullifolius*	across regions	**<0.001**	8.722	48
	Isthme Bas	>0.500	0.073	13
	Australia	0.005	6.716	24
*R. moseleyi*	across regions	**0.001**	4.962	37
	Isthme Bas	0.046	6.673	13
	Australia	0.018	4.818	27
**Quercetins**		**Population**		
		***p*-Value**	***F* Value**	**df Residuals**
*R. biternatus*	across regions	<0.001	17.883	44
	Isthme Bas	0.088	4.697	18
	Australia	**<0.001**	26.440	24
*R. pseudotrullifolius*	across regions	**<0.001**	19.441	46
	Isthme Bas	>0.500	0.202	22
	Australia	**<0.001**	56.282	22
*R. moseleyi*	across regions	**<0.001**	78.570	36
	Isthme Bas	>0.500	1.075	13
	Australia	**<0.001**	147.176	21

* For each species, ANCOVAs were conducted either across or within regions with “population” as factor and environmental conditions as co-variables. Each line represents a separate ANCOVA and only the result for the factor “population” is shown. *p*-values within regions are sequential Bonferroni’s corrected and bold when <0.05.

**Table 2 plants-08-00234-t002:** Differences among populations in the composition of amines or quercetins *.

**Amines**		**Population**		
		***p*-Value**	***F*-Value**	**df Residuals**
*R. biternatus*	across regions	**<0.001**	5.850	44
	Isthme Bas			
	Australia	**<0.001**	9.554	24
*R. pseudotrullifolius*	across regions	**<0.001**	11.256	48
	Isthme Bas			
	Australia	**<0.001**	7.293	24
*R. moseleyi*	across regions	**<0.001**	11.159	48
	Isthme Bas			
	Australia	**<0.001**	18.628	22
**Quercetins**		**Population**		
		***p*-Value**	**F Value**	**df Residuals**
*R. biternatus*	across regions	**<0.001**	7.254	44
	Isthme Bas			
	Australia	**0.001**	16.269	24
*R. pseudotrullifolius*	across regions	**<0.001**	9.495	46
	Isthme Bas			
	Australia	**<0.001**	16.336	22
*R. moseleyi*	across regions	**<0.001**	89.107	36
	Isthme Bas			
	Australia	**<0.001**	126.020	21

* For each species, multivariate ANCOVAS with “population” as factor, environmental variables as covariables, and the levels of all amines (top) or all quercetins (bottom) as dependent variables, done either across regions or within the Australia region (analysis was not possible in Isthme Bas due to collinearity of environmental conditions and populations in this region).

**Table 3 plants-08-00234-t003:** Multivariate relationship between a given environmental variable and compositions of either all amines or all quercetins. Multivariate multiple regression analysis relating a given environmental variable to levels of all compounds *.

**Amines**		**Overall Environment**		**Water Saturation**		**pH**		**Conductivity**		
		***F* Value**	***p*-Value**	***F* Value**	***p*-Value**	***F* Value**	***p*-Value**	***F* Value**	***p*-Value**	**df Residuals**
*R. biternatus*	across regions	4.055	**<0.001**	5.539	**<0.001**	3.022	**0.003**	17.019	**<0.001**	41
	Isthme Bas	4.383	0.361	9.575	0.249	6.470	0.301	2.660	0.452	11
	Australia	5.777	**0.001**	15.488	**<0.001**	2.754	**0.036**	9.360	**<0.001**	13
*R. pseudotrullifolius*	across regions	5.089	**<0.001**	7.616	**<0.001**	2.881	**0.003**	4.837	**<0.001**	40
	Isthme Bas	4.811	**0.008**	17.536	**<0.001**	1.810	0.172	8.924	**<0.001**	10
	Australia	4.221	**0.006**	17.765	**<0.001**	2.866	**0.031**	4.279	**0.006**	13
*R. moseleyi*	across regions	4.006	**<0.001**	1.186	0.334	4.103	**<0.001**	6.392	**<0.001**	24
	Isthme Bas									
	Australia	11.250	**<0.001**	6.576	**0.002**	7.502	**<0.001**	19.503	**<0.001**	11
**Quercetins**		**Overall Environment**								
		***F* Value**	***p*-Value**	***F* Value**	***p*-Value**	***F* Value**	***p*-Value**	***F* Value**	***p*-Value**	**df Residuals**
*R. biternatus*	across regions	4.067	**<0.001**	2.718	**0.013**	2.709	**0.014**	4.779	**<0.001**	43
	Isthme Bas	4.169	**0.010**	3.691	**0.017**	5.254	**0.004**	3.668	**0.017**	17
	Australia	14.812	**<0.001**	105.210	**<0.001**	7.995	**0.002**	10.625	**<0.001**	17
*R. pseudotrullifolius*	across regions	1.139	0.357	5.970	**<0.001**	3.896	**0.001**	2.855	**0.009**	47
	Isthme Bas	22.216	**<0.001**	49.230	**<0.001**	8.963	**<0.001**	40.232	**<0.001**	17
	Australia	3.017	**0.022**	1.714	0.158	7.465	**<0.001**	7.088	**<0.001**	21
*R. moseleyi*	across regions	5.492	**<0.001**	16.485	**<0.001**	10.166	**<0.001**	6.845	**<0.001**	17
	Isthme Bas	126.530	**<0.001**	39.861	**<0.001**	123.590	**<0.001**	3.425	**0.049**	3
	Australia	56.519	**<0.001**	74.673	**<0.001**	40.919	**<0.001**	23.647	**<0.001**	5

* “Overall environment” represents scores along the first axis of an environmental PCA (Principal Component Analysis, Methods). *p*-values are sequential Bonferroni’s corrected and bold when < 0.05. Sample size was insufficient for *R moseleyi* in Isthme Bas.

**Table 4 plants-08-00234-t004:** Multivariate relationship between a given trait and compositions of either all amines or all quercetins. Multivariate multiple regression analysis relating a given trait to levels of all compounds *.

**Amines**		**Phenotype**		**Plant height**		**Number of Leaves**		**Flower Size**		**Leaf Ratio**		**LDMC**		
		***p* Value**	***F***	***p* Value**	***F***	***p* Value**	***F***	***p* Value**	***F***	***p* Value**	***F***	***p* Value**	***F***	**df Residuals**
*R. biternatus*	across regions	0.150	1.491	0.200	1.379	0.540	0.934	0.260	1.272	0.110	1.609	0.130	1.546	41
	Isthme Bas	**0.006**	4.785	0.220	1.590	0.111	2.079	0.150	1.860	**0.005**	5.001	0.300	1.376	11
	Australia	0.350	1.243	0.570	1.041	0.700	0.763	0.230	1.508	0.560	0.455	0.340	1.262	13
*R. pseudotrullifolius*	across regions	**0.006**	2.670	0.720	0.758	0.096	1.664	**0.025**	2.154	0.092	1.680	0.075	1.755	40
	Isthme Bas	**0.005**	5.422	0.600	0.886	**0.008**	4.798	**0.027**	3.418	0.510	1.011	**0.006**	5.173	10
	Australia	**0.012**	3.621	0.760	0.692	**0.005**	4.413	**0.050**	2.515	**0.009**	3.871	**0.006**	4.240	13
*R. moseleyi*	across regions	**0.015**	2.654	0.290	1.271	0.031	2.310	**0.020**	2.516	**0.006**	3.108	0.280	1.288	24
	Isthme Bas													
	Australia	0.088	2.338	**0.006**	5.173	0.140	1.966	0.390	1.209	0.015	4.046	0.280	1.449	11
**Quercetins**		**Phenotype**		**Plant height**		**Number of Leaves**		**Flower Size**		**Leaf Ratio**		**LDMC**		
		***p* Value**	***F***	***p* Value**	***F***	***p* Value**	***F***	***p* Value**	***F***	***p* Value**	***F***	***p* Value**	***F***	**df Residuals**
*R. biternatus*	across regions	**0.025**	2.435	0.240	1.353	**0.024**	2.454	0.320	1.201	0.380	1.105	0.120	1.698	43
	Isthme Bas	0.475	1.004	0.180	1.661	0.084	2.174	0.690	0.713	0.920	0.395	0.910	0.412	17
	Australia	0.350	1.217	0.480	0.997	0.540	0.910	0.680	0.726	0.130	1.878	0.120	1.932	17
*R. pseudotrullifolius*	across regions	**0.005**	3.137	**0.006**	3.053	**0.004**	3.239	**0.006**	3.053	**0.024**	2.424	0.520	0.916	47
	Isthme Bas	**0.006**	4.221	0.240	1.470	**0.024**	3.080	**0.010**	3.780	0.190	1.626	0.570	0.869	17
	Australia	**0.006**	3.822	0.120	1.854	0.072	2.166	0.260	1.383	**0.035**	2.619	0.057	2.311	21
*R. moseleyi*	across regions	**0.006**	4.221	**0.009**	3.869	0.089	2.135	**0.005**	4.384	**0.004**	4.588	0.120	1.932	17
	Isthme Bas	**0.010**	99.388	0.220	3.915	0.183	4.837	**0.006**	166.055	**0.056**	17.241	0.110	8.471	3
	Australia	0.820	0.503	0.460	1.209	0.342	1.606	0.780	0.567	0.250	2.081	**0.006**	19.212	5

* “Phenotype” represents scores along the first axis of a PCA calculated across the traits. Indicated are *F*- and *p*-values after sequential Bonferroni’s correction. *p*-values are sequential Bonferroni’s corrected and bold when < 0.05. Sample size was insufficient for *R. moseleyi* in Isthme Bas.

**Table 5 plants-08-00234-t005:** Effects of region on the relationships between total metabolite contents (amines or quercetins) and (**a**) the environment, or (**b**) the phenotype among populations. The table shows the interaction term between region and either a given environmental variable or a given trait *.

	**Interaction Term between Regions and**											
**Amines**	**(a) Overall Environment**			**Soil Water Saturation**			**pH**			**Conductivity**		
	***p*-Value**	***F***	**R^2^**	***p*-Value**	***F***	**R^2^**	***p*-Value**	***F***	**R^2^**	***p*-Value**	***F***	**R^2^**
*R. biternatus*	0.059	3.723	0.213	0.157	2.07	0.116	0.291	1.138	0.007	0.608	0.267	0.52
*R. pseudotrullifolius*	0.349	0.892	0.306	**0.023**	5.467	0.336	0.952	0.004	0.309	0.116	2.55	0.32
*R. moseleyi*	0.111	2.647	0.527	**<0.001**	23.706	0.317	0.068	3.522	0.375	**0.051**	4.048	0.55
**Quercetins**	**Overall Environment**			**Soil Water Saturation**			**pH**			**Conductivity**		
	***p*-Value**	***F***	**R^2^**	***p*-Value**	***F***	**R^2^**	***p*-Value**	***F***	**R^2^**	***p*-Value**	***F***	**R^2^**
*R. biternatus*	0.949	0.004	0.21	0.191	1.755	0.196	0.104	2.747	0.356	0.073	3.351	0.18
*R. pseudotrullifolius*	**0.005**	8.719	0.104	**0.005**	8.719	0.011	**<0.001**	14.71	0.204	0.312	1.044	0.17
*R. moseleyi*	0.593	0.291	0.006	0.502	0.459	0.314	0.574	0.321	0.062	**0.001**	11.71	0.31
	**Interaction Term between Regions and**											
**Amines**	**(b) Phenotype**			**Plant height**			**Number of Leaves**					
	***p*-Value**	***F***	**R^2^**	***p*-Value**	***F***	**R^2^**	***p*-Value**	***F***	**R^2^**			
*R. biternatus*	0.91	0	0.042	0.89	0.774	0.025	0.094	4.709	0.066			
*R. pseudotrullifolius*	0.99	0.029	0.26	0.31	0.203	0.27	**0.001**	10.82	0.39			
*R. moseleyi*	**0.046**	9.388	0.35	**0.042**	5.76	0.32	0.41	0.048	0.26			
**Quercetins**	**Phenotype**			**Plant height**			**Number of Leaves**					
	***p*-value**	***F***	**R^2^**	***p*-value**	***F***	**R^2^**	***p*-value**	***F***	**R^2^**			
*R. biternatus*	0.63	0.005	0.008	0.76	0.757	0.022	0.61	0.152	0.17			
*R. pseudotrullifolius*	0.77	0.084	0.025	0.87	0.026	0.1	**0.022**	7.952	0.13			
*R. moseleyi*	0.17	3.349	0.18	0.08	0.078	0.11	0.57	2.624	0.1			

* Interaction terms are taken from multiple regression analyses that contain also the respective raw variables. “Overall environment” and “Phenotype” represent scores along the first axis of an environmental or morphological PCA, respectively (Methods) calculated across the “soil water saturation”, “pH” and “conductivity”. *F* and sequential-Bonferroni corrected *p*-value (bold if < 0.05) of the interaction term; and adjusted r squared of the multiple regression. Sample sizes for amines/quercetines: *R. biternatus* = 58/52; *R. pseudotrullifolius* = 57/48; *R. moseleyi* = 46/26.

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
