# Peer review of "Variations of Secondary Metabolites among Natural Populations of Sub-Antarctic Ranunculus Species Suggest Functional Redundancy and Versatility"

_plants, 2019, doi:10.3390/plants8070234_

Round 1
Reviewer 1 Report
It is not clear how do the authors have been able to characterize the compounds (amine sand quercetins). Overall, the chemical analysis needs to be detailed in order to judge its relevance.
Minor concern, figure 2 is difficult to read.
The conclusion is interesting if the authors can support with more evidences the compound ID.
Author Response
Response to Reviewer 1 comments.
Point 1:
It is not clear how do the authors have been able to characterize the compounds (amine sand quercetins). Overall, the chemical analysis needs to be detailed in order to judge its relevance.
Response 1:
Analysis of amines:
We added details on the identification and quantitation methods we used in the HPLC analysis of amines (namely lines 501-509 and 515-522). This included the complete list of external standards, details on the initial column calibration processes, their periodic injection to verify the stability of retention times and quantitation curves throughout the set of analyses, and their occasional use to confirm compound identity in case of doubt by spiking the sample with known amounts as is the rule in HPLC analysis.
Analysis of quercetins:
Details were added lines 531 to 545.
Point 2:
Minor concern, figure 2 is difficult to read.
Response 2:
We redid this Figure, now Figure 4.
Point 3:
The conclusion is interesting if the authors can support with more evidences the compound ID.
Response 3:
Please see Response 1.
Reviewer 2 Report
The manuscript presented by Labarrere et al focus on metabolite profiling in sub-antarctic Ranunculus species. The authors investigate patterns of variation in natural populations in two sampling locations with similar environmental variables. The experimental design is strong and the conclusions well supported by the data. Although not a native English speaker, I think that the English langage deserves improvements.
However, the present version has to be revised in some aspects. Please see my recommendations below.
The result part is quite difficult to read because the authors mention significance in different comparisons (eg L146). I suggest the authors to better describe what are is really compared in each situation or to find another way of presenting the data. Maybe small conclusion sentences should be added at the end of each paragraph (eg L202 : « hence, plant height was only moderately correlated to quercetin content »). Concerning data presentation, heatmaps in the format populations x traits may be envisaged to summarize tables (especially Table5). To my opinion, data presentation is the main weakness in this manuscript. While the approach and the results are scientifically sound, it is difficult to get the main message from the way the data are presented. In addition, please use Fig and Table numbers in the discussion (eg L320 : where the reader can find the most appropriated results supporting this observation?)
I think that the use of Kerguelen Islands will be more appropriated throughout the text.
In tables 3, 4 and 5, the authors used PCA for dimensionality reduction and stated that component 1 represented phenotype or overall environment. But as it can be clearly observed in Fig1 top panel, component 1 which represents the highest % of initial variability but not necessarily all the variables. As a consequence, the authors should provide an indication of the best correlated variables with component 1 so that the reader may be aware of traits that are truly represented by this component. Please provide % of variance for both components in Fig1.
Concerning the sampling procedure, it may be preferable to indicate if metabolite content changed all over the 6 week sampling periode. A simple ANOVA should indicate if the sampling periode has a potential influence on metabolite content (rather that citing unpublished data).
L24 : use flavonol rather than quercetins
L28 : within the same environment
L28 : it is not clear what lies behind « same trait »
L45 : pollinator attraction
L59 : have examined
L66 : remove « not always », the structure is awkward . Change to eg , High diversity of secondary metabolites has already been reported to correlate with functional diversity.
L79 : would be to search for
Table1 : replace commas with « . » decimal separator
L140 : collinearity
L277 : remove the comma after « individual »
L331 : Concerning environmental conditions
L344 : I do not understand how same vegetation types support that other environmental variables may not have been missed. Similar vegetations may have underwent similar adaptative evolution to another, not controlled, environmental variable (such as enlightment/wind?).
L371 : This is also the case for quercetins.
L383-384 : I disagree with that sentence. Amines and flavonols arise from different amino-acids and no conversion between amines and flavonols has been describe (which is probably impossible from a chemical point of view). Please describe this part more accurately.
L449 : what does 1100hours stand for ?
L490 : please indicate standard supplier
L529 : please provide m/z
L551 : there is no reference number for R
p { margin-bottom: 0.25cm; line-height: 115%; }Author Response
Response to Reviewer 2 comments.
Please note, all our changes are in red font in the manuscript.
Point 1:
The result part is quite difficult to read because the authors mention significance in different
comparisons (eg L146). I suggest the authors to better describe what are is really compared in each
situation or to find another way of presenting the data. Maybe small conclusion sentences should be
added at the end of each paragraph (eg L202 : « hence, plant height was only moderately correlated
to quercetin content »). Concerning data presentation, heatmaps in the format populations x traits
may be envisaged to summarize tables (especially Table5). To my opinion, data presentation is the
main weakness in this manuscript. While the approach and the results are scientifically sound, it is difficult to get the main message from the way the data are presented. In addition, please use Fig and Table numbers in the discussion (eg L320 : where the reader can find the most appropriated results supporting this observation?)
Response 1:
We added 2 Figures (1 and 2) to better illustrate the results.
We added Figure and Table numbers in the discussion at the appropriate places, as was suggested by the reviewer.
In addition, we provided the raw data of plant amine and quercetin compositions in supplementary material (Tables S3 and S4, respectively).
Point 2:
I think that the use of Kerguelen Islands will be more appropriated throughout the text.
Response 2:
Iles Kerguelen is an official name. We use to use it in our publications (e.g. Lehnebach et al. 2017, J. Biogeogr), so, if possible we had rather keep this name.
Point 3:
In tables 3, 4 and 5, the authors used PCA for dimensionality reduction and stated that
component 1 represented phenotype or overall environment. But as it can be clearly observed in Fig1
top panel, component 1 which represents the highest % of initial variability but not necessarily all the
variables. As a consequence, the authors should provide an indication of the best correlated variables
with component 1 so that the reader may be aware of traits that are truly represented by this
component.
Response 3:
We now explain in the legends:
“”We also used an integrative “overall environment” variable calculated as the scores along the first axis of a PCA across the individual environmental variables. This axis was most strongly correlated to water saturation in R. biternatus, and to pH in the two other species. To statistically test whether relationships between compositions of metabolites and the morphological phenotypes differ between regions we took the same approach as for environment, replacing environmental by morphological variables. Again we identified an integrative “overall phenotype” variable calculated as the scores along the first axis of a PCA across the individual morphological variables. This axis was most strongly correlated to Plant height in all three species and to leave number in R. biternatus, and to LDMC in the other two species.”
Please provide % of variance for both components in Fig1.
To our knowledge there is no explained variance associated to axes of an RDA. However we now provide the explained variance for the entire analyses: “R² are 0.49, 0.73, ad 0.54 for Isthme Bas, Australia, and across-region, respectively.”
Point 4:
Concerning the sampling procedure, it may be preferable to indicate if metabolite content changed all over the 6 week sampling periode. A simple ANOVA should indicate if the sampling periode has a potential influence on metabolite content (rather that citing unpublished data).
Response 4:
We now explain: “Because metabolite composition may vary within a given plant at a given moment between leaves of different developmental stages [49, 51], we sampled an appropriate and constant leaf developmental stage, i.e., fully developed photosynthetic leaves as in previous work [52]. We moreover verified whether inclusion of sampling month into an ANOVA relating region to the different compounds reduced the number of significant relationships between region and compounds and found that this was not the case. For amines even the identity of the affected compounds remained the same in 15 out of 20 cases.”
L24 : use flavonol rather than quercetins
Done
L28 : within the same environment
We believe this was a misunderstanding of our meaning by the reviewer. We changed the writing of the sentence to : “In several cases, a given metabolite showed different or even opposite relations with a same environmental condition or a same trait”.
L28 : it is not clear what lies behind « same trait »
Please see above correction.
L45 : pollinator attraction
Done
L59 : have examined
Done
L66 : remove « not always », the structure is awkward . Change to eg, High diversity of
secondary metabolites has already been reported to correlate with functional diversity.
Done
L79 : would be to search for
Done
Table1 : replace commas with « . » decimal separator
Done
L140 : collinearity
Done
L277 : remove the comma after « individual »
Done
L331 : Concerning environmental conditions
This was our mistake, as this paragraph is about morphological traits. We removed these words.
L344 : I do not understand how same vegetation types support that other environmental
variables may not have been missed. Similar vegetations may have underwent similar adaptative
evolution to another, not controlled, environmental variable (such as enlightment/wind?).
We agree with this statement. In fact, our previous work strongly supports the environmental conditions we chose as being discriminant for the compositions of our species. So, we changed our text to: “Our results might possibly suggest that we have left out important environmental variables. However, we consider this unlikely given that our previous work supports the soil hydric variables we used as discriminant for the compositions in respectively amines [26] and quercetins [39] of our study species”.
L371 : This is also the case for quercetins.
We agree, and we say so in the next sentence: “Such is also true of quercetins”, so we did not change anything here.
L383-384 : I disagree with that sentence. Amines and flavonols arise from different amino-acids
and no conversion between amines and flavonols has been describe (which is probably impossible
from a chemical point of view). Please describe this part more accurately.
We are sorry, this resulted from a wrong writing. We changed “Both in amine and quercetin families, the compounds share metabolic pathways and may derive one from each other [7, 38]” to : “Within each of the amine and quercetin families, the compounds share metabolic pathways and may derive one from each other [7, 38]”.
L449 : what does 1100hours stand for ?
11h – we changed this writing to “11 hours and 17 hours”.
L490 : please indicate standard supplier
We included these informations at first mention of the internal standard, diaminoheptane, and after the added list of external standards.
L529 : please provide m/z
This was done in the new redaction of the methods for flavonol analyses.
L551 : there is no reference number for R
We added this reference.
Reviewer 3 Report
The authors of the present article delivered a relevant manuscript which in my view should be published in Plants. However, major adaptations are needed before publication. In its current form the results fully rely on statistics, which in my view is an incomplete presentation of the results. To gain credibility, clarity and substantiation to the readers of Plants, it is absolutely necessary to include good, representative insights of the raw experimental data next to the statistical figures.
Pg.2/3, Line 91-106. Even though the authors decided to focus on only two known subclasses of secondary metabolites (amines and quercetins) there are numerous other well-known subclasses that may be influenced by environmental factors or traits (SCFA, (poly)saccharides, nucleotides, etc.). In the current introduction the authors did not elaborate on them. Can the authors comment on the fact that other metabolite classes were not considered? And for what reason the authors limited their study by measuring only the already known metabolites? In other words, what is the novelty of this research at this point? Hence, it could have been a good opportunity to explore new (unknown) metabolites.
Pg.3, Section 2. Without providing any insights in the experimental data, the first results were directly presented in form of statistical tables. ANCOVA (and even multivariate ANCOVA) is rather complex and not trivial for the main audience, and therefore need to be introduced, explained and discussed clearly (this is also missing in Section 4.8). To get some feeling with the data and the methodology, as well as the typical response differences (effect sizes) across populations and regions it is necessary to provide the readers with some representative examples of the experimental output. Given this experimental output, the interpretation of the statistical figures will become more intuitive.
Pg.3, Line 123. ANCOVA was applied on the between- and within regions. This seems not a logical way of analyzing the data. What was the rationale behind this choice, i.e. why not using 2-way ANCOVA?
Pg.3, Line 123. It is common practice to include the equation of the variance model that will be fitted. This model equation is now missing and will provide good insight in the sources of variation (and its interactions) that are investigated in the present study.
Pg.3, Line 124. At this point it is unknown what the environmental conditions (covariates) are? This important information is now a bit hidden, and firstly provided in the caption text of Table 2.
Pg.3, Table 1. Some effects were rather strong (p<0.001) and may be visualized in a separate plot to demonstrate/confirm the population differences in the raw experimental data. For example, a x-y plot representing the amine and quercetin concentrations of R. moseleyi, showing the differences among populations across regions.
Pg.4, Line 128-130; Pg. 4, Line 137-142; Pg. 5, Line 160-166; Pg.6, Line 193-198; Pg.7, Line 213-219. Captions of Tables 1-5 needs to be more carefully explained/included in the method section of the main body text (Section 4.8), including the “Error df” terminology (Tables 1 and 3), the connection to the PCA analysis (Table 3), Bonferroni’s correction (Table 4), and the introduction of interaction terms (Table 5). The comments are overall far too complex, and too important, to include it into a caption. This particularly counts for the caption of Table 3.
Pg.4, Line 145-146; Pg.5, Line 167-169. For clarification reasons it may be good to plot examples of the relationships between environmental conditions and amine/quercetin content. This will provide direct (visual) insights in the effect sizes of the experimental data.
Figure 1. Quality of Figure is extremely poor. Sample labels are too small, as well as the labels of the PCA plot axis. What is the explained variance of these PC’s? How is the data pre-processed before the actual PCA analysis (include in 4.8)? This information is currently missing in the manuscript.
Author Response
Response to Reviewer 3 comments.
Please note, all our changes are in red font in the manuscript.
The authors of the present article delivered a relevant manuscript which in my view should be
published in Plants. However, major adaptations are needed before publication. In its current form
the results fully rely on statistics, which in my view is an incomplete presentation of the results. To
gain credibility, clarity and substantiation to the readers of Plants, it is absolutely necessary to include
good, representative insights of the raw experimental data next to the statistical figures.
Point 1:
Pg.2/3, Line 91-106. Even though the authors decided to focus on only two known subclasses of
secondary metabolites (amines and quercetins) there are numerous other well-known subclasses that
may be influenced by environmental factors or traits (SCFA, (poly)saccharides, nucleotides, etc.). In
the current introduction the authors did not elaborate on them. Can the authors comment on the fact
that other metabolite classes were not considered? And for what reason the authors limited their
study by measuring only the already known metabolites? In other words, what is the novelty of this
research at this point? Hence, it could have been a good opportunity to explore new (unknown)
metabolites.
Response 1 :
We agree that other metabolites may be influenced by environmental factors or traits. To address the complex variation of plant metabolite composition across populations, environments, traits, and regions in natura, we needed to target on metabolites known (and which we showed) to vary in relation to environments or traits. Due to the small size of some of our species, we were also limited by the amount of material collected in the field. Untargeted approaches may be used in the future to address variation at larger scales. We changed the text to better justify our choices:
“This also requires metabolites that are known to vary and play roles in the response to environment or in traits. Among other metabolites, such is the case with amines and flavonoids [34, 35]. Previous work comparing the amine metabolomes of 9 species showed that amine composition of populations of plants in Iles Kerguelen varied in relation to both species and the environment [26]. Furthermore, previous work on flavonoids in the three Ranunculus species growing in Iles Kerguelen showed that quercetins were the only flavonols in these species and that composition of populations varied in relation to both species and the environment [39].(…) Therefore, to address the complex variation of plant metabolite composition across populations, environments, traits, and regions in natura, we performed targeted analyses of amines and quercetins. Using two independent metabolite families aimed at reinforcing our conclusions.”
Point 2:
Pg.3, Section 2. Without providing any insights in the experimental data, the first results were directly presented in form of statistical tables. ANCOVA (and even multivariate ANCOVA) is rather complex and not trivial for the main audience, and therefore need to be introduced, explained and discussed clearly (this is also missing in Section 4.8). To get some feeling with the data and the methodology, as well as the typical response differences (effect sizes) across populations and regions it is necessary to provide the readers with some representative examples of the experimental output. Given this experimental output, the interpretation of the statistical figures will become more intuitive.
Response 2:
We added the raw data of plant amine and quercetin compositions in supplementary material (Tables S3 and S4, respectively).
We illustrated the underlying patterns across populations. Illustrating the more complex ANCOVA that accounts in addition for environmental covariables, would require presenting some residual values, i.e. not the “experimental data” that the referee required. As the conclusions between the complex and the simple model are the same we preferred presenting the simple one.
Point 3:
Pg.3, Line 123. ANCOVA was applied on the between- and within regions. This seems not a logical way of analyzing the data. What was the rationale behind this choice, i.e. why not using 2- way ANCOVA?
Response 3:
We now explain: “We analyzed the difference between populations, accounting for environmental conditions: Total amine content in sample = pH(continuous) + Conductivity(continuous) + Water saturation(continuous) + population(6 categories) + error. We consistently repeated this procedure across all populations and across only the populations within a given region to explore whether populations are more different in one region or another or whether populations vary only across regions, and not between.”
Point 4:
Pg.3, Line 123. It is common practice to include the equation of the variance model that will be fitted. This model equation is now missing and will provide good insight in the sources of variation (and its interactions) that are investigated in the present study.
Response 4:
We now explain the model “Total amine content in sample = pH(continuous) + Conductivity(continuous) + Water saturation(continuous) + population(6 categories) + error”;
Point 5:
Pg.3, Table 1. Some effects were rather strong (p<0.001) and may be visualized in a separate plot to demonstrate/confirm the population differences in the raw experimental data. For example, a x-y plot representing the amine and quercetin concentrations of R. moseleyi, showing the differences among populations across regions.
Response 5:
This seems to be the same comment as the one “Pg3, Section 2” and we responded above. We also added an illustration of Results in Table 5. See above.
Point 6:
Pg.4, Line 128-130; Pg. 4, Line 137-142; Pg. 5, Line 160-166; Pg.6, Line 193-198; Pg.7, Line 213-219.
Captions of Tables 1-5 needs to be more carefully explained/included in the method section of the
main body text (Section 4.8), including the “Error df” terminology (Tables 1 and 3), the connection to
the PCA analysis (Table 3), Bonferroni’s correction (Table 4), and the introduction of interaction terms
(Table 5). The comments are overall far too complex, and too important, to include it into a caption.
This particularly counts for the caption of Table 3.
Response 6:
We shortened below-table legends by half for all tables. Writing is more stringent now, obvious terms are not explained anymore and PCA definitions have been evacuated to the Methods.
Point 7:
Pg.4, Line 145-146; Pg.5, Line 167-169. For clarification reasons it may be good to plot examples
of the relationships between environmental conditions and amine/quercetin content. This will
provide direct (visual) insights in the effect sizes of the experimental data.
Response 7:
We included Figure 2 to illustrate the particularly interesting, interacting effects of region and environment on total amines and quercetins.
Point 8:
Figure 1. Quality of Figure is extremely poor. Sample labels are too small, as well as the labels
of the PCA plot axis. What is the explained variance of these PC’s? How is the data pre-processed
before the actual PCA analysis (include in 4.8)? This information is currently missing in the
manuscript.
Response 8:
We improved the resolution of the figures.
Round 2
Reviewer 3 Report
Accepted after comments and adaptations made by authors.